# Half-quantized layer hall effect as a probe of quantized axion field

Jiayuan Hu[1,8], Binbin Wang[1,8], Humian Zhou[2,8], Tongtong Jia[3], Zheng Sun[3], Chang Liu[3], Bo Zhang[3], Dong Qian ®[3], Tingxin Li ®[3], X. C. Xie[2,4,5], Yunchuan Kong[1], Chui-Zhen Chen ®[6,7] ✉ & Di Xiao ®[1] ✉

The axion insulator is a topological phase characterized by a quantized axion field, which manifests as a half-quantized anomalous Hall conductance (AHC) localized on individual surface layers. Experimental realization of this effect has remained elusive because opposite surface contributions typically cancel. We engineered magnetic axion insulator heterostructures using molecular beam epitaxy. By positioning the Fermi level asymmetrically, we isolated one topological surface state within the magnetic gap while displacing the other into a metallic regime. This technique reliably produced a half-quantized layer-resolved AHC of $e^2/2h$, termed the half-quantized layer Hall effect (LHE), across both parallel and antiparallel magnetization configurations in over ten devices. Our findings provide direct electrical evidence of the half-quantized LHE, a boundary signature of the quantized axion field in the bulk, which resolves a key challenge in the experimental verification of this topological quantum state and establishes a framework for spatially engineering quantized topological response.

Axions were originally proposed in high-energy physics to resolve the strong CP violation problem in quantum chromodynamics (QCD), but have remained experimentally elusive[1]. In condensed matter systems, axion insulators, characterized by a quantized axion field, offer a promising platform to experimentally explore axion-like physics[2–4]. This quantized axion field gives rise to a unique topological magnetoelectric (TME) response, originating from a quantized topological term in the electromagnetic action[2], $\Delta\mathcal{L} = \theta(\alpha/4\pi^2)\mathbf{E} \cdot \mathbf{B}$, where $\theta = \pi$ is the quantized axion angle, and $\alpha \equiv e^2/\hbar c$ is the fine-structure constant. However, direct verification of the TME effect by measuring the quantized responses to external fields is still a challenging open task[5]. This motivates the search for a definitive experimental signature of quantized axion field beyond the bulk TME response in axion insulators.

Surface electronic transport provides a boundary signature of the quantized axion field in the bulk. When time-reversal symmetry is locally broken at the surface (e.g., by an out-of-plane magnetization), the resulting gapped surface states induce a 2D Chern-Simons term that generates a half-quantized anomalous Hall conductance (AHC) $\sigma_{xy} = \pm e^2/2h$[6]. This half-quantized effect and the TME effect are two inseparable physical consequences linked by the bulk-boundary correspondence. Despite its theoretical elegance, the axion insulator phase in transport experiments presents an enigmatic scenario[7–9]: antiparallel magnetized surfaces yield vanishing net Hall response, obscuring direct access to the critical $e^2/2h$ per-surface quantization. Current experimental paradigms rely on indirect bulk transport signatures – zero Hall plateaus coexisting with insulating behavior – which remains phenomenologically ambiguous against trivial band

[1]Huawei Technologies Co. Ltd., Shanghai, China. [2]International Center for Quantum Materials, School of Physics, Peking University, Beijing, China. [3]State Key Laboratory of Micro-nano Engineering Science, School of Physics and Astronomy, Shanghai Jiao Tong University, Shanghai, China. [4]Interdisciplinary Center for Theoretical Physics and Information Sciences, Fudan University, Shanghai, China. [5]Hefei National Laboratory, Hefei, China. [6]School of Physical Science and Technology, Soochow University, Suzhou, China. [7]Institute for Advanced Study, Soochow University, Suzhou, China. [8]These authors contributed equally: Jiayuan Hu, Binbin Wang, Humian Zhou. ✉e-mail: czchen@suda.edu.cn; xiaodi12@hisilicon.com

insulators[10]. Fortunately, recent advances in layer Hall effect (LHE) physics suggest a pathway forward: spatial separation of Berry curvature generates a layer-resolved Hall effect[11–15]. The direct observation of layer-resolved half-quantization consequently stands as the pivotal requirement for unambiguous verification of the axion insulator phase.

However, these layer-resolved Hall responses have thus far remained unquantized in axion insulator candidate MnBi$_2$Te$_4$[11,16,17], due to the influence of non-topological effects, rendering the experimental evidence of a quantized response elusive. Specifically, the existing implementations in exfoliated MnBi$_2$Te$_4$ flakes suffer from intrinsic limitations: (i) strict space-time $PT$ symmetry that limit parameter tunability, (ii) micron-scale device sizes introducing fabrication challenges and possible finite-size effects, and (iii) complex interlayer antiferromagnetic ordering that obscures the surface Hall contributions[18–20]. Even under applied symmetry-breaking electric fields, these constraints fundamentally compromise the extraction of intrinsic quantized axion responses[11,16], let alone half-quantized LHE.

Here, we overcome these challenges through precision synthesis of MBE-grown magnetic TI heterostructures (Fig. 1a), engineered as prototypical axion insulators with atomically controlled magnetic interfaces. Differential V/Cr doping[21,22] across top and bottom surface layers enables robust anti-parallel magnetization configuration, or the axion configuration[2]. Previous studies mainly focus on the axion insulating regime with Fermi levels positioned within dual magnetic gaps $|M_{(t/b)}|$ (Fig. 1b), where meron-like spin textures in each surface band generate counter-propagating layer-dependent persistent half-quantized edge currents[23]. Unlike the quantized chiral edge states in a Chern insulator, these edge currents emerges as chiral edge channels when they coexist with the metallic surface state, carrying half-quantized Hall conductances $\sigma_{xy}^{t,b} = \text{sign}(M_{t/b})e^2/2h$ in transport measurement[24–26]. Nonetheless, critical parameters governing the surface-specific responses remain underexplored[15] - particularly the exchange field tunability, the layer thickness control, as well as the layer-dependent chemical potential engineering. This synthetic control over competing energy scales enables unprecedented isolation of layer-resolved quantum transport signatures, as demonstrated through our surface-selective transport measurements.

## Results

We synthesized a series of magnetic TI heterostructures via MBE on annealed SrTiO$_3$ (111) substrates (see Methods), which serve as bottom gate dielectrics. The prototypical axion insulator heterostructure comprises three distinct layers: 3 quintuple layers (QL) of Cr$_x$(Bi,Sb)$_{2-x}$Te$_3$ (CBST), $m$ QL of undoped (Bi,Sb)$_2$Te$_3$ (BST) spacer, and 3 QL of V$_y$(Bi,Sb)$_{2-y}$Te$_3$ (VBST) (denoted as CBV, Fig. 1c), where $x$ and $y$ represent magnetic doping concentration. The chemical potential of the sample can be adjusted via changing the Bi:Sb ratio, $\eta$. For electrical transport characterization, devices were fabricated through two approaches: bottom-gated Hall bars are mechanically defined using tungsten tip scribing; dual-gated Hall bar devices are patterned via photolithography with 50 nm AlO$_x$ top dielectric. Low-temperature transport measurements were performed in a dilution refrigerator (base temperature $T = 30$ mK, unless other specified) with four-terminal lock-in techniques. For a CBV Device A1 ($m = 20$, $x = 0.19$, $y = 0.11$, $\eta = 0.83$), the overall magneto-transport behavior is as expected at the charge neutrality point (CNP) with the bottom gate voltage $V_{bg}$=150 V. Figure 1d demonstrates the magnetic-field-driven phase transitions between Chern insulator ($\sigma_{xy} = \pm e^2/h$) and axion insulator ($\sigma_{xy} = 0$), with parallel and anti-parallel magnetization configurations respectively. The coercive field is $H_{c1} = 0.76$ T for the top VBST layer and $H_{c2} = 0.12$ T for the bottom CBST layer. The midpoint of zero Hall plateau locates near $\mu_0 H = \pm 0.4$ T.

To reveal layer-resolved quantum transport, we systematically probe the asymmetric Fermi level configuration by tuning the bottom gate voltage $V_{bg}$ from CNP toward hole-doping regimes. As $V_{bg}$ decreases below CNP, the Hall conductance hysteresis evolves from a two-step transition to a single half-quantized plateau (Fig. 2a). At zero magnetic field, the antiparallel magnetization configuration exhibits $\sigma_{xy}$ growth from 0 to $e^2/2h$ while the parallel configuration shows $\sigma_{xy}$ reduction from $e^2/h$ to $e^2/2h$ (Fig. 2b). Remarkably, both configurations ultimately converge to the same half-quantized value. This convergence demonstrates complete suppression of the bottom layer's Hall conductance as the Fermi level penetrates its massive Dirac band, well below the magnetic gap (Fig. 2c), leaving only the top layer's half-quantized AHC. This observation demonstrates the precise quantized form of the layer Hall effect[11,12,15], thereby termed as the half-quantized layer Hall effect (HQLHE). The HQLHE constitutes the central discovery of this work. Crucially, it exhibits two defining characteristics that distinguish it from conventional LHE phenomena in PT symmetric systems: (i) identical quantization $\sigma_{xy} = \pm e^2/2h$ persists across both parallel and antiparallel magnetic states; (ii) quantization largely maintains even without external electric field[15] (e.g. $V_{bg} = 0$ V in Fig. 2b).

Two critical factors enable this half-quantized phenomenon. Firstly, the Cr doping level is deliberately kept relatively low with

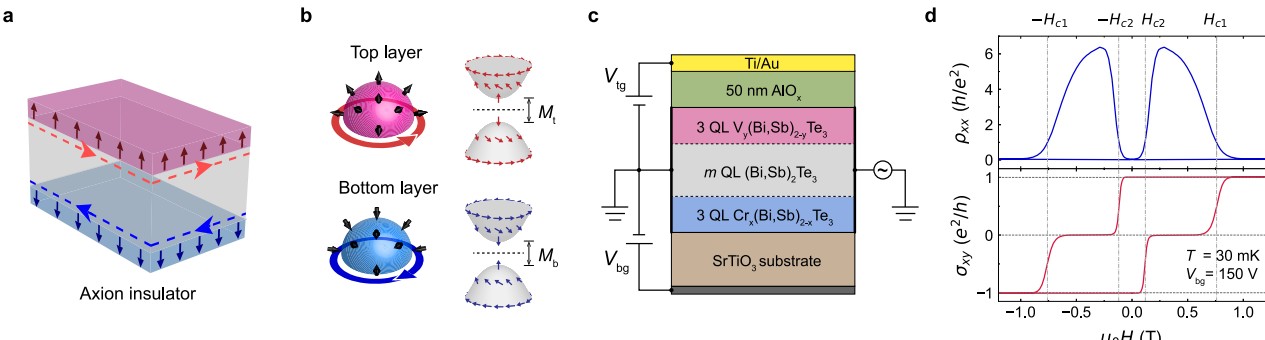

**Fig. 1 | Axion insulator in magnetic topological insulator heterostructures.**
**a** Schematic of the axion insulator in a sandwich magnetic TI heterostructure with antiparallel magnetization configuration. The red and blue dashed arrows circulating on the side surface illustrate the Hall currents from top and bottom surfaces, respectively. **b** Illustration of the half-quantized surface Hall effect in the axion insulator. The massive Dirac band of the top (bottom) surface layer has a meron-like spin texture, with spins polarized outward (inward). When the Fermi level lies inside the magnetic gaps, net Hall conductance of half-quantized Hall effect from two surfaces exactly cancels each other. **c** Schematic drawing of the dual-gated devices. $V_{bg}$, bottom gate voltage; $V_{tg}$ top gate voltage. **d** Longitudinal resistance $\rho_{xx}$ and Hall conductance $\sigma_{xy}$ as a function of magnetic field ($\mu_0 H$) of Device A1 at CNP $V_{bg} = 150$ V.

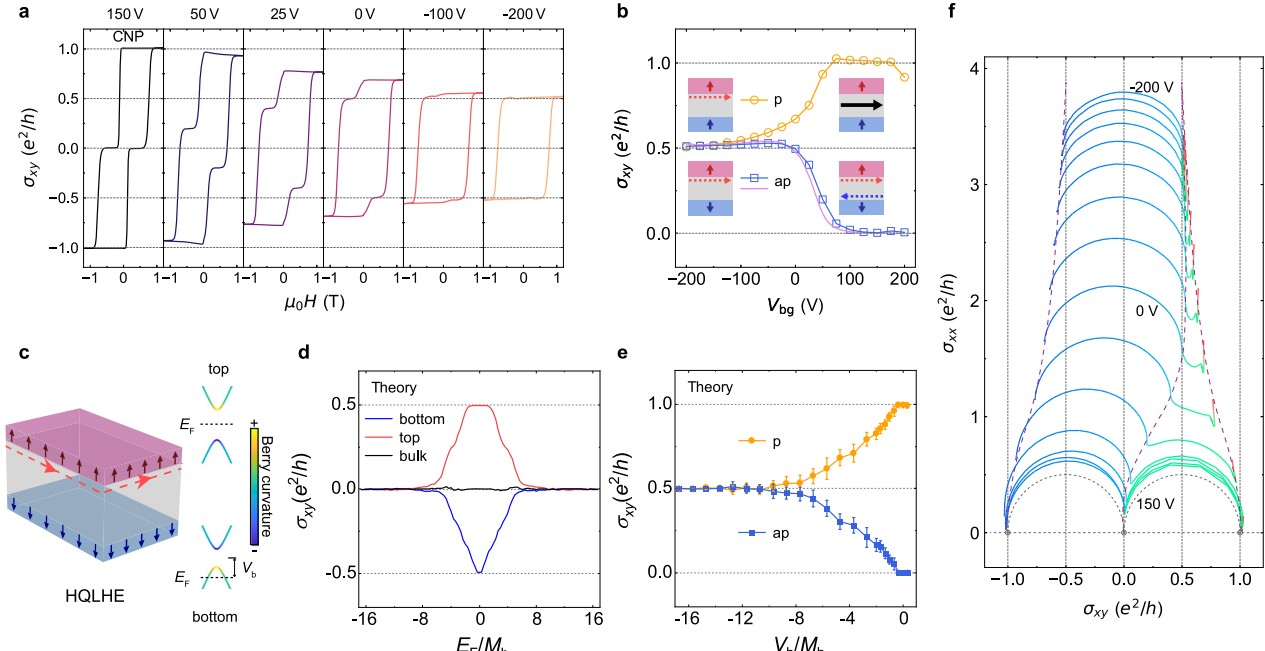

**Fig. 2 | Observation of the half-quantized layer Hall effect in Device A1.**
**a** Dependence of the Hall conductance $\sigma_{xy}$ on the magnetic field $\mu_0 H$, measured at different bottom gate voltage $V_{bg}$. **b** The dependence of the Hall conductance on $V_{bg}$ with parallel (p, orange at zero field) and antiparallel (ap, blue at $\pm 0.4$ T, pink at zero field) magnetization configuration. The schematics denotes the 4 main states, i.e. insulating/hole-doped bottom surface with parallel/antiparallel magnetization configurations. The red and blue dashed arrows on the side surface illustrate the Hall currents from top and bottom surfaces, respectively. The black arrow indicates the chiral edge state of the Chern insulator. **c** Schematic of the HQLHE in a sandwich magnetic TI heterostructure with antiparallel magnetization configuration. Berry curvature concentrated near the band edges. When the Fermi level at bottom

surface is tuned deep into the hole-doped regime by $V_b$, the Berry curvature vanishes, leaving only the single-surface Hall current (dashed red arrows). **d** Numerical calculations of the layer-resolved AHC $\sigma_{xy}^t$ (red) and $\sigma_{xy}^b$ (purple) as a function of fermi energy ($E_F$). The bulk Hall signal (black) is vanishingly small throughout the whole range of $E_F$. **e** Numerical results of the total $\sigma_{xy}$ at zero field as a function of $V_b$ with parallel/antiparallel magnetization configurations. Error bars correspond to the standard deviation obtained from 30 independent disorder configurations. **f** Renormalization group flow in $(\sigma_{xy}, \sigma_{xx})$ plane with varying $V_{bg}$ ($-200$ V to 150 V) and $\mu_0 H$ ($\pm 1.2$ T loop as in **a**). The purple dashed lines are guide to the eye for the flows of the fixed points from $e^2/h(\pm 1.0, 0)$ and $(0, 0)$ to points along $\sigma_{xy} = \pm e^2/2h$.

$x = 0.19$ to ensure that the Berry curvature from the bottom surface band is concentrated near the gap edge (Fig. 2c). At $V_{bg} = -200$ V, the band of bottom layer is partially filled and exhibits nearly zero Berry curvature, contributing negligibly to the total Hall conductance. In comparison, it is challenging to realize HQLHE in a device with high Cr doping level (see Supplementary Fig. 1). Secondly, the high density of bulk trap states in $(Bi,Sb)_2Te_3$ effectively screens electric fields[27,28]. Thereby, the Fermi level remains within the top surface magnetic gap when tuning $V_{bg}$, preserving the top layer's half-quantized Hall conductance.

To fully understand the HQLHE theoretically, we construct a four-band effective 3D TI Hamiltonian with the axion configuration by introducing Zeeman splitting $H_M = M_{t/b} s_z$ on the top and bottom surfaces. We apply an additional potential energy term $H_G = V(z)$ with the value $V_t$ ($V_b$) on the top (bottom) surface, to independently control the surface Fermi energy of the top and bottom layer. We also incorporate the random onsite potentials to include disorder effects that are inevitable in real devices and then evaluate the layer-dependent AHC by a real-space Kubo formula[29] (more details in Methods). Numerical results in Fig. 2d show that the layer-resolved AHC $\sigma_{xy}^t$ and $\sigma_{xy}^b$ are half-quantized when Fermi level $E_F$ falls within the magnetic gap but decay to zero at higher energies. By electrostatic gating of the bottom surface potential $V_b$, the total AHC $\sigma_{xy}$ can be continuously tuned from zero (axion insulator) or $e^2/h$ (Chern insulator) to $e^2/2h$ (HQLHE) when $E_F$ lies within the top surface's magnetic gap (see Fig. 2e and Supplementary Fig. 2a). The consistence between the theoretical simulations and experiments results unambiguously demonstrates the half-quantized nature of layer-resolved hidden Berry curvature in axion insulator heterostructures.

To further elucidate the universality of the HQLHE, we construct a renormalization group flow diagram (RGFD) by mapping magneto-transport trajectories onto the $(\sigma_{xy}, \sigma_{xx})$ conductivity plane[30] (Fig. 2f). At CNP ($V_{bg} = 150$ V), the system traverses two semicircular arcs with radii $e^2/2h$ under $\mu_0 H$ variation, connecting three stable fixed points: $(\pm 1.0, 0)e^2/h$ for the Chern insulator phase and $(0, 0)$ for the axion insulator phase[31]. Upon tuning $V_{bg}$ to $-200$ V, the RGFD undergoes a fundamental transformation: the right semicircle collapse into points along $\sigma_{xy} = e^2/2h$, while the left semicircle shifts its origin from $(-0.5, 0)e^2/h$ to $(0, \sigma_{xx})$. The finite longitudinal conductance $\sigma_{xx}$ increases up to about 3.5 $e^2/h$, signaling metallic transport. This RGFD demonstrates a continuous quantum phase transition from conventional integer quantum Hall fixed points to the emergent HQLHE metallic state[32]. The 1/2-shift of $\sigma_{xy}$ reflects the scaling law of single-flavor Dirac fermions. Crucially, the smooth connection between these phases on the same device provides irrefutable evidence that the $(0,0)$ fixed point comes from the axion insulator state rather than trivial insulator state, conclusively eliminating alternative explanations such as surface-state hybridization[10]. The RGFD evolution further established HQLHE as the boundary signature of quantized axion field in magnetic TIs.

We demonstrate tunable HQLHE by engineering layer-resolved Berry curvature through spacer thickness, surface-specific chemical potentials, exchange fields, and thermal fluctuations. Systematic thickness-dependent studies reveal robust HQLHE signatures across devices (Supplementary Fig. 3), confirming its intrinsic surface origin. At $m = 5$ with anti-parallel magnetization, we observe slightly reduced $\sigma_{xy} = 0.48 \, e^2/h$ at $V_{bg} = -200$ V and $\mu_0 H = 0$ T, along with further reduced $\sigma_{xy}$ at $\mu_0 H = -0.4$ T. This indicates strengthened surface

hybridization and interlayer magnetic coupling effect due to the reduced spacer thickness. At $m = 30$, the AHC converges to $\sigma_{xy} = 0.55\ e^2/h$ at $V_{bg} = -200$ V, slightly exceeding the expected half-quantization. Noticeably, another type of quantization deviation appears near the QAH plateau, such as the AHC peak at $V_{bg} = 150$ V in Supplementary Fig. 3h. We attribute these two types of quantization deviations to the quasi-1D non-chiral edge conduction channels[33], as supported by theoretical modeling[34] (see detailed discussion in Supplementary Note xii). For HQLHE, the optimal quantization accuracy is achieved under two conditions: vanishing AHC from the deep hole doped surface, and reduced conductance from the parasitic non-chiral edge channels. Among all tested devices, devices with spacer thickness $10 \le m \le 20$ show the most accurate AHC of $\sigma_{xy} = 0.505 \pm 0.019\ e^2/h$ despite other possible errors (See Supplementary Fig. 5 and detailed discussion in Supplementary Notes x, xii, and xiii). On the other hand, at the insulating QAH state, $\sigma_{xy}$ is most accurate at CNP where both surface states and quasi-1D helical edge states vanish.

Utilizing dual-gated Hall bar devices (Fig. 1c), we achieve the manipulation of the HQLHE through independent chemical potential control of top and bottom surfaces. Figure 3a-d maps the Hall conductance landscape of Device B1 across parallel and antiparallel magnetization configurations. In parallel magnetization Fig. 3a, the QAH phase ($\sigma_{xy} = e^2/h$, light green region, inside the red dashed lines) spans the entire top-gate operational range, demonstrating stronger chemical potential tunability at the bottom surface. The QAH phase turns into the axion insulator phase (dark blue region inside the red dashed lines) in Fig. 3b with antiparallel configurations. Noticeably, the AHC $\sigma_{xy}$ of both configurations converges to $\sigma_{xy} \approx 0.45\ e^2/h$ when $V_{bg} < 0$ (plateaus in lower halves of Fig. 3a and c), as in the line plots in Fig. b

and d. We attribute the deviation of AHC from half-quantization to sample quality degradation during fabrication, which possibly caused the actual width of the device several microns smaller than the original design. These AHC plateaus also exhibit top gate voltage ($V_{tg}$) dependent modulation, with $\sigma_{xy}$ decreasing systematically as $V_{tg}$ increasing.

To interpret the dual-gated experimental observations, we perform theoretical calculations by independently controlling the Fermi energies of the top and bottom layers (Fig. 3e). The calculated AHC inside area indicated by the pink dashed lines, with slight tilt due to inevitable tuning of top/bottom gate on the opposite surfaces, qualitatively fits the data in Fig. 3c. Furthermore, a precisely quantized layer Hall conductance, controllable via a dual-gate setup, is clearly visible in Fig. 3e. Although full access to both p- and n-type HQLHE regimes remains limited by current gating efficiency, emergent same-sign layer Hall signals on both sides of $\sigma_{xy} \approx 0$ at $V_{tg} > 0$ (e.g., $V_{tg} = 15$ V in Fig. 3d) reveals a distinct mechanism from conventional electric-field-induced antisymmetric LHE with respect to $E_F = 0$[15]. Additional results of specially adjusted CBV devices (Supplementary Fig. 6) where the bottom surface is highly n-doped, demonstrate a continuous transition of Hall conductance from $e^2/h$ towards $e^2/2h$ in the n-doped regime. This complements the p-doped data and confirms the symmetric nature of the effect, distinct from previous studies[11]. We thus argue that by further enhancing the device fabrication quality and gating efficiency, one could in principle access a broader LHE phase diagram, with 4 HQLHE plateaus (2 plateaus for each $\pm e^2/2h$) in certain parameter space (Supplementary Fig. 7).

To further confirm the universality of the HQLHE in magnetic TIs, it is essential to directly verify the half-quantized Hall conductance of

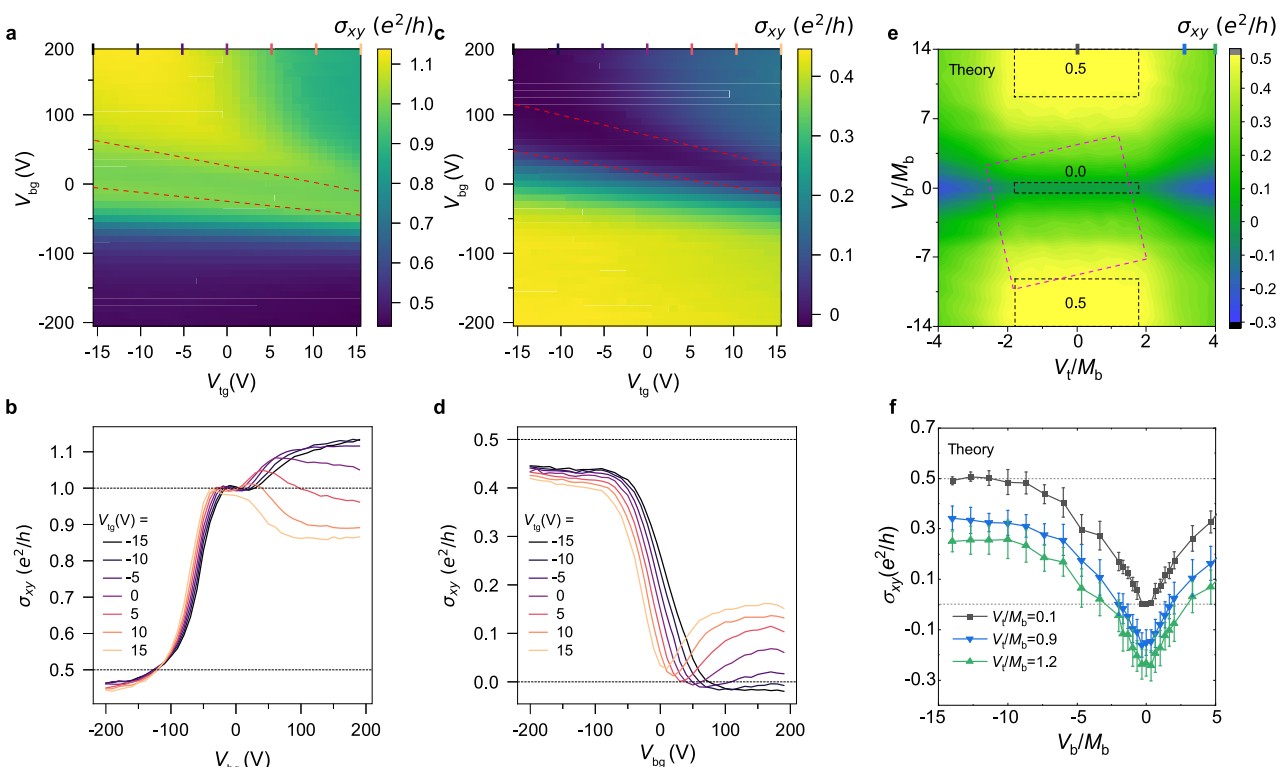

**Fig. 3 | Manipulation of the HQLHE in a dual-gated Device B1. a, c** Zero field Hall conductance $\sigma_{xy}$ as a function of $V_{tg}$ and $V_{bg}$ with parallel (**a**) and antiparallel (**c**) magnetization configuration. The regions indicated by red dashed lines correspond to (**a**) QAH ($\sigma_{xy} = e^2/h$) and (**c**) axion insulator ($\sigma_{xy} = 0$) plateaus respectively. **b, d** Zero field Hall conductance $\sigma_{xy}$ as a function of $V_{bg}$ at various $V_{tg}$ with parallel (**b**) and antiparallel (**d**) magnetization configuration, extracted from (**a**) and (**c**).

**e** Calculated phase diagram for our axion insulator 4-band model, with $\sigma_{xy}$ as a function of $V_t/M_b$ and $V_b/M_b$. The regions defined by the black dashed lines correspond to HQLHE ($\sigma_{xy} = e^2/2h$) and axion insulator ($\sigma_{xy} = 0$) plateaus. The region defined by the pink dashed lines is a simulation of (**c**). **f** Extracted $\sigma_{xy}$ as a function of $V_b/M_b$ at 3 different $V_t/M_b$ values. Error bars correspond to the standard deviation obtained from 30 independent disorder configurations.

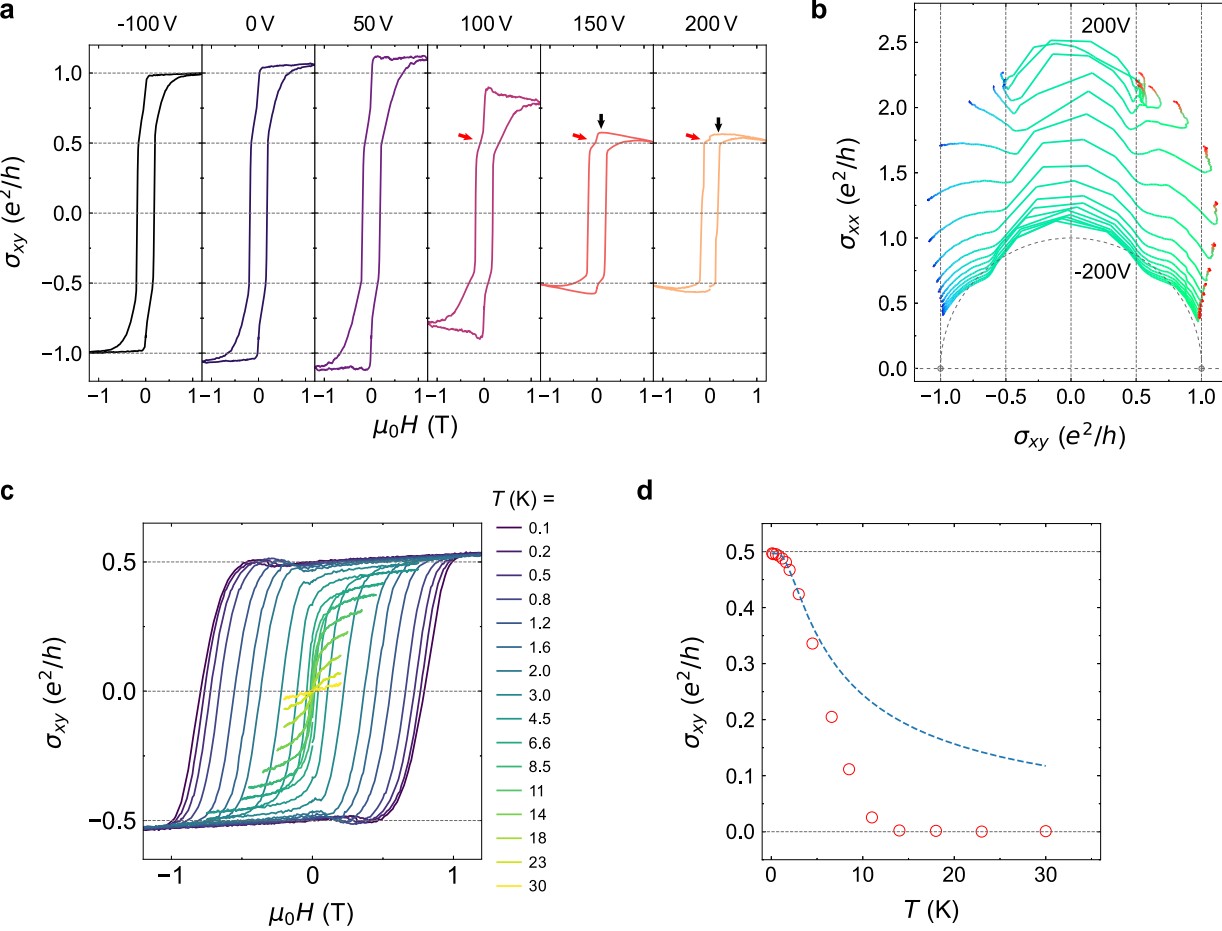

**Fig. 4 | Manipulation of the HQLHE with modified exchange fields and elevated temperature. a** Dependence of the Hall conductance $\sigma_{xy}$ on the magnetic field $\mu_0 H$, measured at different bottom gate voltage $V_{bg}$ for a VBC Device C1. The black arrows indicate excessive Hall conductance from the VBST layer due to insufficient gating. The red arrows denote the kinks near half quantized $\sigma_{xy} = e^2/2h$. **b** Renormalization group flow in $\left(\sigma_{xy}, \sigma_{xx}\right)$ plane with varying $V_{bg}$ (−200 V to 200 V) and $\mu_0 H$ (±1.2 T loop as in **a**). The purple dashed lines are guide to the eye for the flows of the fixed points from $e^2/h(\pm 1.0, 0)$ to points along $\sigma_{xy} = \pm e^2/2h$. **c** Hall conductance loops of Device A2 as a function temperature at a fixed bottom gate voltage $V_{bg} = -200$ V. **d** Temperature dependence of the AHC at zero magnetic field. The blue dashed line is the fit of the data at temperatures $\leq 3$ K following the method discussed in Supplementary Note viii.

both the V-doped and Cr-doped surface layers. To this end, we engineer Device C1 with inverted doping architecture (VBC configuration: V-doped bottom layer, Cr-doped top layer) and reduced vanadium concentration ($y = 0.03$). This structural modification enhances chemical potential tunability of the VBST layer via gating through the STO substrate. The lowered V doping results in a mix state of weak ferromagnetism and superparamagnetism[35,36], precluding the axion antiparallel magnetization (see Fig. 4a). Despite this constraint, HQLHE persists in the parallel magnetization configuration (see also Supplementary Fig. 8 for 2 examples of CBC devices), demonstrating the half-quantized Hall conductance of Cr-doped surface layers.

As shown in Fig. 4a, gate modulation towards $V_{bg} = 200$ V drives a continuous transition from the QAH state to the HQLHE regime (see numerical simulations in Supplementary Fig. 2b), which proves the vanishing Berry curvature mechanism also applies in the electron doped regime, as expected theoretically. The small overshoot of $\sigma_{xy}$ indicated by black arrows is attributed to the residual Berry curvature of VBST layer because of insufficient gating. Emergent kinks near $\sigma_{xy} = e^2/2h$ indicated by red arrows, reflects layer-selective quantization from the CBST top surface layers. By changing $V_{bg}$, the RGFD in Fig. 4b reveals different behaviors compared to Device A1: while the

conventional axion insulator fixed point $(\sigma_{xy}, \sigma_{xx}) = (0, 0)$ disappears, there is also a continuous evolution from QAH-like semi-circle[37] (radius $e^2/h$) to HQLHE trajectories (radius $e^2/2h$) with finite $\sigma_{xx}$.

Finally, the temperature dependence provides critical insights into the HQLHE. Since the insulating gapped surface provides the half-quantized Hall conductance, the temperature stability of HQLHE directly reflects the interplay between thermal energy $k_B T$ and exchange gaps $\Delta$ of the insulating surface. Figure 4c illustrates this relationship through temperature-dependent AHC measurements on Device A2 (nominally the same parameters with Device A1), where the HQLHE remains above $0.47\ e^2/h$ up to about 2 K. Conservative analysis[38] of the $\sigma_{xy}(T)$ scaling below the Curie temperature $T_C$ yields a minimum estimates of exchange gap ($\Delta_{VBST} > 0.7$ meV) for the V-doped surface (Fig. 4d), which is compatible with the $T_C \sim 14$ K. This temperature analysis indicates that HQLHE seems to be simply limited by the exchange field strength of the insulating gapped surface. Notice that there are variations of the estimated exchange gap $\Delta$, even with nominally the same doping concentration (See Supplementary Fig. 9 for additional temperature dependence data). We attribute this inconsistency to the different Fermi level positions inside the actual exchange gaps of VBST layers. (see Supplementary Note viii for detailed discussions.)

## Discussion

The observations of the HQLHE suggest potential exciting developments ahead. First, in an ideal realization, dual-gate control could spatially redistribute half quantized Hall currents between top and bottom surfaces - a capability with profound implications for developing quantized layer-selective transport. This represents a significant advance beyond the non-quantized, layer-polarized transport reported by Gao et al.[11,12] and non-tunable half-quantization in the semi-magnetic system by Mogi et al.[39]. Achieving the dynamic control over layer selectivity positions the magnetic axion insulator as a compelling platform for topological layertronics[40,41]. Second, the discovery of the HQLHE provides a boundary signature of the quantized axion field in a magnetic topological insulator with antiparallel magnetization configuration, offering a new platform for the systematic study of the axion electrodynamics. Crucially, the axion insulator with the HQLHE fundamentally differs from the semi-metallic parity anomaly state in semi-magnetic TI systems[38,39,42], which clearly lacks a well-defined axion field. In HQLHE, time-reversal symmetry (TRS) is broken locally on both surfaces, in contrast to parity anomaly states where TRS is preserved on one surface. This dual-surface symmetry breaking not only activates the layer degree of freedom, but also allows continuous tuning between QAH/axion phases and HQLHE phases merely through chemical potential engineering.

To summarize, our systematic investigation of magnetic TI heterostructures reveals the HQLHE as a robust boundary signature of the quantized axion field. Through careful tuning of surface hybridization, surface chemical potentials, and magnetic configurations, we demonstrate half-quantization of layer-resolved Hall conductance $\sigma_{xy} = \pm e^2/2h$ across a dozen of devices. These findings provide strong experimental evidence supporting the long-predicted connection between layer-resolved Berry curvature and the layer-selective quantum transport in magnetic three-dimensional TIs. This work bridges topological quantum matter, field theory, and quantum metrology, opens a rich landscape for future developments in layertronics, non-linear Hall effect[43,44], spintronics[45–47], and quantum information processing applications[48,49].

## Methods

### MBE growth

The magnetic doping $(Bi,Sb)_2Te_3$ topological insulator(TI) sandwich structures were fabricated by using a commercial MBE system (Fermion MBE-300) with a base vacuum better than $1.5 \times 10^{-10}$ mbar. The insulating 0.5 mm $SrTiO_3$ (111) substrates used for growth were initially etched with 1% hydrofluoric acid (HF) solution for 2 minutes and then annealed in a tube furnace at 980 °C under flowing oxygen for 3 h (1st thermal annealing). Subsequently, the substrates were etched again with deionized (DI) water (resistivity > 18 MΩ · cm) at 80 °C for 1.5 h and then annealed at 980 °C under $O_2$ atmosphere for 3 h (2nd thermal annealing). Through this two-step thermal-annealing procedure, a passivated and atomically flat surface of $SrTiO_3$ (111) substrates was obtained. After that, the heat-treated $SrTiO_3$ (111) substrates were outgassed at 600 °C for 10 min. Next, high purity Sb (99.99999%), Bi (99.9999%), Te (99.99999%), Cr (99.999%), V (99.99%) were evaporated from Kundsen effusion cells. The chromium (Cr)/vanadium (V) doping concentration $x/y$ was precisely controlled through the evaporation temperature of Cr/V cell, and the value of $x/y$ was calibrated by X-ray photoelectron spectroscopy (XPS) shortly after calibration growths (see Supplementary Note xii). The substrate was held at 260 °C during the growth of all heterostructures. The flux ratio of Te per (Bi + Sb + Cr /V) was set to greater than 20 to prevent Te deficiency in the films. The Bi/Sb ratio was optimized to tune the chemical potential of the magnetic axion insulator heterostructures near the charge neutral point. The heterostructure growth process was monitored in situ by reflection high-energy electron diffraction (RHEED), where the high crystal quality was confirmed by streaky and sharp 1×1 patterns. The growth rate of TI films is ~0.4 QL per minute, calibrated by RHEED oscillations and thickness measurements using an atomic force microscopy (AFM).

### Hall bar device fabrication

The SrTiO3 substrates were used as the gate-dielectric to fine-tune the chemical potential of the films. For the bottom gated devices, the MBE films were mechanically scratched into six-terminal Hall bars by a tungsten needle controlled by a programmed probe station. The geometry specifications were measured using an optical microscope imaging system (see examples in Supplementary Note x). For the dual gate devices, the Hall bar patterns were defined via standard photo lithography (AZ1500 series resist) and subsequent argon plasma etching. Ti (3 nm)/Au (35 nm) electrodes were deposited onto the cleaned area of the Hall bar terminals, followed by a liftoff process. Atomic layer deposition (ALD) of 50 nm $AlO_x$ dielectric layer was conducted at 100 °C. Finally, a Ti (3 nm)/Au (35 nm) top gate was fabricated on the effective area of the Hall bar device using a second photo lithography and liftoff process. The highest temperature during device fabrication was kept below 110 °C to minimize potential degradation. The ohmic contacts were made by soldering indium dots onto the previously defined electrodes. For $AlO_x$ covered electrodes, scratching off the insulating dielectric material prior to soldering guaranteed good ohmic contact.

### Electrical transport measurement

The magneto-electric transport was conducted in a Physical Property Measurement System (PPMS, Quantum Design DynaCool, for temperature range 1.7 K and above) and a dilution refrigerator (Oxford Instruments, Triton500, for temperature range 30 mK and above). The magnetic field was applied perpendicular to the sample plane. Longitudinal and Hall voltages were measured simultaneously using standard lock-in techniques at a low frequency around 7Hz (SRS SR830 lock-in amplifier). The gate voltages were applied using Keithley 2400 or 2614 B source meters. The excitation current was set to 10 nA, balanced for a good signal to noise ratio and minimized heating effect. Two major types of measurements were conducted as following: (i) sweeping the magnetic field at fixed gate voltages, and (ii) sweeping the gate voltages with known magnetization configurations (parallel or anti-parallel) at zero external magnetic field. Although the Hall bar geometry of the scratched sample looked symmetric under microscope, the inevitable slight misorientation or errors leads to the mixing of $V_{xx}$ and $V_{yx}$ signals. The calculation of the sheet conductance $\sigma_{xx}$ and $\sigma_{xy}$ followed the tensor relations $\sigma_{xx} = \rho_{xx}/(\rho_{xx}^2 + \rho_{yx}^2)$ and $\sigma_{xy} = \rho_{yx}/(\rho_{xx}^2 + \rho_{yx}^2)$, where $\rho_{xx}$ and $\rho_{yx}$ data obtained through conventional symmetrization and anti-symmetrization procedures, respectively. For a comparison between raw and processed data, see examples in Supplementary Note xi and xvi. Information of all the measured devices in this study is summarized in Supplementary Table 2.

### Theoretical models of axion insulator heterostructure and calculations of layer-resolved Hall conductance

we consider a 3 D TI with antiparallel magnetization alignment surfaces [see Fig. 1(a)] that is realized in our experiment. The four-band effective Hamiltonian is $H = H_0 + H_M$, where

$$H_0(\mathbf{k}) = \sum_{i=x,y,z} Ak_i\sigma_x \otimes s_i + \left(M_0 - Bk^2\right)\sigma_z \otimes s_0 \tag{1}$$

describes the 3D TI[50], with model parameters $A$, $B$, and $M_0$. $\sigma_i$ and $s_i$ are Pauli matrices for the orbital and spin degrees of freedom, respectively. We discretize the Hamiltonian H into $N_x \times N_y \times N_z$ lattice sites. $H_M = M(z)\sigma_0 \otimes s_z$ is the Zeeman splitting. $M(z) = M_b$ for $z=1$ (bottom surface), $M(z) = M_t$ for $z = N_z$ (top surface), and $M(z) = 0$ for $1 < z < N_z$.

We introduce $H_G = V_G(z)\sigma_0 \otimes s_0$ to independently control the surface Fermi energy of the top and bottom layer, which is realized by applying voltages on the top and bottom gates in our experiment. $V_G(z) = V_t$ for $z = 1, 2$, $V_G(z) = V_b$ for $z = N_z, N_z - 1$, and $V_G(z) = 0$ for $2 < z < N_z - 1$. We incorporate the random disorder as $H_D = V_D(\mathbf{r})\sigma_0 \otimes s_0$, where the $V_D(\mathbf{r})$ is uniformly distributed within $[-W/2, W/2]$ with the disorder strength $W$.

We discretize the Hamiltonian $H$ on lattice sites:

$$H = \left[ \sum_{\mathbf{i}} \mathbf{c}_{\mathbf{i}}^\dagger \mathcal{M}_0 \mathbf{c}_{\mathbf{i}} + \left( \sum_{\mathbf{i}, j=x,y,z} \mathbf{c}_{\mathbf{i}}^\dagger \mathcal{T}_j \mathbf{c}_{\mathbf{i}+\hat{\mathbf{e}}_j} + \text{h.c.} \right) \right] + \sum_{\mathbf{i}} \mathbf{c}_{\mathbf{i}}^\dagger \mathcal{M}_z \mathbf{c}_{\mathbf{i}} \quad (2)$$

where $\mathcal{M}_0 = \left( M_0 - 2B/a_x^2 - 2B/a_y^2 - 2B/a_z^2 \right) \sigma_z \otimes s_0$, and $\mathcal{T}_j = B/a_j^2 \sigma_z \otimes s_0 - iA/\left(2a_j\right) \sigma_x \otimes s_j$, and $\mathcal{M}_z = M(z)\sigma_0 \otimes s_z$. Here, $a_j = a$ is the lattice constant. $\mathbf{c}_{\mathbf{i}}$ ($\mathbf{c}_{\mathbf{i}}^\dagger$) is the annihilation (creation) operator at site $\mathbf{i}$, and $\hat{\mathbf{e}}_j$ is a unit vector in the j direction for $j = x, y, z$. The sample thickness is given by $d = N_z a_z$. The model parameters for numerical calculations are fixed as $A/a = 1.0, B/a^2 = 0.5, M_0 = 1.0, W = 2.5$, and $N_x \times N_y \times N_z = 25 \times 25 \times 6$. $M_b = 0.3, M_t = 0.6$ in the Fig. 2 in the main text. Error bar correspond to the standard deviation obtained from 30 independent disorder configurations.

The layer-dependent Hall conductance is calculated by using the real-space Kubo formula[29]:

$$\sigma_{xy}(z) = \frac{2\pi i e^2}{h} \text{Tr}\left\{ P\left[ -i[\hat{x}, P], -i[\hat{y}, P] \right] \right\}_z \quad (3)$$

with periodic boundary condition in both the $x$ and $y$ direction. $(\hat{x}, \hat{y})$ denotes the position operator, and $\text{Tr}\{...\}_z$ is trace over the wave functions of the $z$th layer. $P$ is the projector onto the occupied states of the effective Hamiltonian $H$. The bottom surface Hall conductance is $\sigma_{xy}^b = \sum_{z=1}^{z=2} \sigma_{xy}(z)$, the top surface Hall conductance is $\sigma_{xy}^t = \sum_{z=N_z-1}^{z=N_z} \sigma_{xy}(z)$, and the bulk Hall conductance is $\sigma_{xy}^{bulk} = \sum_{z=3}^{z=N_z-2} \sigma_{xy}(z)$.

The six-layer tight-binding model is a standard and computationally efficient effective model used to capture the essential low-energy physics of the two surface states of a 3D topological insulator. For a film thickness of 20 QLs, the system is firmly in the 3D limit where the hybridization between top and bottom surfaces is negligible. In this regime, the surfaces are electronically independent, and their behavior is well-described by the effective model. The model is not intended to reproduce the bulk band structure of the 20 QL film but to correctly capture the qualitative behavior of the surface states under gating.

## Data availability

The data used in this study are openly available at [https://doi.org/10.5281/zenodo.17773455].

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

## Acknowledgements

We thank J. Xu for supporting this research project and insightful discussions. We thank C. Zhou for his help with low temperature facilities. We thank Y.-H. Li for valuable discussions on the theoretical analysis. We also thank Z. Zeng, B. Wang, J. Chen, B. Wu, A. Zhang for their help with sample preparation and characterization facilities. Our appreciation also goes to Huawei Semiconductor Failure Analysis Center Platform and Nanofab at Suzhou Institute of Nano-Tech and Nano-Bionics, CAS for offering us a platform of device fabrication and characterization. C.-Z.C. acknowledges support from the National Key R&D Program of China Grant (2022YFA1403700) and the Natural Science Foundation of Jiangsu Province Grant (BK20230066). H.-M.Z. is supported by the National Natural Science Foundation of China (Grants No.123B2058). X.-C.X. acknowledges support from the Innovation Program for Quantum Science and Technology (grant no. 2021ZD0302400). T.L. acknowledges support from the National Natural Science Foundation of China (Nos. 12350403, 92265102), the Natural Science Foundation of Shanghai (Nos. 24QA2703700, 24LZ1401100) and the New Cornerstone Science Foundation through the XPLORER PRIZE. D.Q. acknowledges support from the National Key R&D Program of China (No. 2022YFA1402400).

## Author contributions

D.X. conceived and designed the experiment. J.H. grew the magnetic TI samples with help from B.Z. and D.Q. B.W. prepared the bottom gated devices and performed the dilution transport measurements. B.Z. performed the TEM measurements. With help from T.L., T.J., Z.S. and C.L. prepared the dual gated devices and carried out the PPMS transport measurements. H.Z. and C.-Z.C. did all calculations and provided theoretical support, with help from X.C.X. D.X., J.H., B.W., H.Z. and C.-Z.C. analyzed the data and wrote the manuscript, with input from all authors. Y.K. managed and oversaw the research project. D.X. was responsible for the overall direction, planning and integration among research units.

## Competing interests

The authors declare no competing interests.
