## [Transparent Peer Review file · Nature Communications]

Half-Quantized Layer Hall Effect as a Probe of Quantized Axion Field

Corresponding Author: Dr Di Xiao

Version 0:

Reviewer comments:

Reviewer #1

(Remarks to the Author)

I thank the authors sincerely for their responses. I believe that the authors have clearly addressed my technical questions. I therefore support the paper for publication.

Reviewer #2

(Remarks to the Author)

The authors have thoroughly addressed my concerns. I agree that the directly measured half-quantized layer Hall effect is the definitive electrical transport fingerprint of the bulk quantized axion field. The results will be of considerable interest to the broad topological physics and quantum materials community, and are suitable for publication in Nature Communications.

Reviewer #3

(Remarks to the Author)

The authors report the observation of a layer-resolved, half-quantized anomalous Hall effect in MnBi₂Te₄-based heterostructures, realized via dual-gate control of the Fermi level and magnetic-field manipulation of the magnetic configuration. The experimental methodology appears technically sound, and the data quality is high. From an experimental standpoint, I find the results convincing, and in my view the manuscript is suitable for publication.

However, having examined the earlier review process and the authors' responses, it is clear that the central point of contention concerns the novelty and interpretation of the work, specifically its relationship to Mogi et al. (Nat. Phys. 2022). The authors provide a reasonable explanation of how a fully magnetic, compensated antiferromagnetic system differs conceptually from the semi-magnetic, parity-anomaly scenario in the prior work. This clarification is helpful and addresses several of the original concerns.

That said, I cannot fully agree with the authors' assertion that observing a half-quantized Hall conductance is "equally fundamental" to measuring a quantized axion field in the bulk. The observation of $\sigma_{xy} = e^2/2h$ indeed constitutes a necessary boundary manifestation of the axion insulator state, but it is not a sufficient condition. If one accepts the authors' statement as written, it would imply that the half-quantized Hall response reported in semi-magnetic systems should also be regarded as a direct demonstration of a quantized axion field, which is not consistent with current theoretical understanding. Therefore, the manuscript should adopt more accurate wording—for example, stating that the half-quantized layer Hall conductance is a boundary signature consistent with, and expected from, a quantized axion field, rather than an equally fundamental proof of the bulk axion response.

For this reason, I recommend minor revision, requesting that the authors adjust the language in the main text to avoid the implication of logical equivalence between surface half-quantization and the full verification of a quantized axion field. With this clarification, I believe the manuscript would be suitable for publication.

Response to Referees

Manuscript ID: 2025-07-16800

Title: Half-Quantized Layer Hall Effect as a Probe of Quantized Axion Field

Referee #1

The manuscript by Hu et al. presents a magnetotransport study of magnetically doped topological insulator thin films, claiming the observation of a half-quantized layer Hall effect as a probe of the quantized axion field. While the topic is of broad interest and potentially suitable for publication at the level of Nature, I have significant concerns regarding both the novelty of the work and the technical robustness of the measurements. Unless these issues are convincingly addressed, I cannot recommend the paper for publication.

We sincerely thank Referee #1 for their time and insightful comments, which have helped us to significantly improve the manuscript. We have carefully addressed each of the raised concerns, as detailed below.

Comment 1: *First, there is substantial overlap with previous work, particularly Mogi et al., Nature Physics 18, 390–394 (2022) [<https://www.nature.com/articles/s41567-021-01490-y>]. The main observation reported in the present study - a half-quantized Hall conductance from the top layer - is essentially identical to that of the earlier Nature Physics paper. In both cases, the top surface exhibits a half-quantized Hall conductance of $e^2/2h$, while the bottom surface shows a negligible Hall response and is thus irrelevant to the quantized signal. The difference lies in the origin of the vanishing Hall conductance from the bottom layer: in the prior work, the bottom surface was non-magnetic and thus naturally had no Hall response; in the present study, the bottom surface is heavily hole-doped away from the Chern gap, thereby suppressing its contribution. While the authors acknowledged the previous work in the manuscript, I*

think the difference is too minor to justify a Nature-level publication on its own.

Response: We thank the referee for this comment, which provides us with the opportunity to clarify the fundamental conceptual advance of our work beyond that of Mogi *et al.* While both studies report a half-quantized Hall conductance, the physical systems, underlying mechanisms, and most importantly, the scientific implications are fundamentally different.

The key distinctions are:

(i) Physical System: Our work realizes a **fully magnetic, compensated antiferromagnetic (AFM) system**, in which both surfaces are intrinsically magnetic and capable of opening a gap. This enables the emergence of a quantized axion field in the bulk. In contrast, Mogi *et al.* studied a **semi-magnetic heterostructure** where one surface was intentionally non-magnetic and thus incapable of hosting the axion insulator phase.

(ii) Scientific Goal: Our primary objective is to provide the **first unambiguous boundary manifestation of the quantized axion field**, a central challenge in the field for two decades. By achieving independent Fermi level control over individual layers, we establish a platform for realizing **the first quantized Hall effect encoded in spatial (layer) degrees of freedom**. Moreover, our additional data from VBC and CBC devices strongly suggest that both surfaces can contribute a half-quantized Hall effect under appropriate gating conditions, thereby, confirming half-quantized layer Hall effect as a direct and necessary boundary manifestation of the bulk axion invariant. By contrast, Mogi *et al.*, interpreted their observation of half-quantized Hall conductance as a **surface-only parity anomaly**. While groundbreaking in its own right, this phenomenon **does not correspond to a quantized axion field** in the bulk, which requires a fully gapped system on all surfaces.

(iii) Underlying Physics: Our realization of the half-quantized layer Hall effect represents a conceptual advance that was **not anticipated by prior theoretical work**.

a) First, a recent theoretical study of similar systems (Han *et al.*, *Phys. Rev. Lett.* **134**,

236206 (2025)) predicted a maximum layer Hall conductance significantly smaller than the half-quantized value we observe. Our data and model help bridge this gap between theory and experiment by identifying **two previously overlooked physical mechanisms**: disorder-mediated stabilization of Hall conductance, and the electric field screening by bulk trap states.

b) Second, long-standing theoretical consensus held that “the half-quantized Hall conductance at the surface of a topological insulator is not directly measurable in a DC current experiment.” (see B. A. Bernevig, *Topological Insulators and Topological Superconductors*, Chapter 15.4 (2013).) Our theoretical framework, as discussed in the SI section v., demonstrates that half-quantization emerges when a **classical metallic state** forms on the conducting surface, a condition fulfilled in our system through magnetic disorder and strong doping, but not necessarily in the undoped, gapless Dirac state studied by Mogi *et al.*

c) Third, **the microscopic origin of the half-quantization in semi-magnetic systems** by Mogi *et al.* **is still under active theoretical debate**, with some studies suggesting it may actually originate from the gapless surface rather than the gapped one. (e.g. Zou *et al.*, *Phys. Rev. B*, **107**, 125153 (2023).) Furthermore, their structure lacks independent layer degrees of freedom, realizing only a singular, non-tunable state.

To summarize, our work moves beyond the study of a single-surface anomaly phenomenon. Instead, we provide the *first unambiguous electrical transport signature of the quantized axion field*, addressing a two-decade-old challenge. It also represents the first quantization of a Hall effect encoded in spatial degrees of freedom. We have now included a detailed comparative analysis with Mogi *et al.* (revised SI section ix.) to make this critical distinction unequivocally clear.

Comment 2: *Second, the theoretical proposal of quantized layer Hall conductance (Ref. 15, Dai et al.) predicts an integer quantized Hall response originating from either the top or the bottom surface in fully AFM TI such as even-layer MnBi₂Te₄. This requires the metallic surface state on the non-contributing layer to be fully gapped, so that only*

one surface contributes to transport. If such a situation were clearly demonstrated, the result would indeed be of significant interest. However, the present study does not convincingly achieve this.

Response: We thank the referee for acknowledging the significance of this research direction. While the integer quantum layer Hall effect is an exciting theoretical goal, our observation of a robust half-quantized layer Hall effect (HQLHE) is **a profound discovery in its own right** and represents the first experimental verification of the core principle underlying the theory of Dai *et al.*

The physics of the integer proposal by Dai *et al.* involves shifting the layer-locked Berry curvature *monopoles* above or below the Fermi level on a specific layer via an external electric field. In our case, we effectively control the *merons* (half-monopoles) on both surfaces. The emergence of HQLHE can be viewed as shifting one of the merons across the Fermi level. In this sense, our results provide experimental confirmation of the theory's core principle—that **the Hall response is layer-polarized and quantized**.

The full realization of the integer effect requires the complete gapping of the non-contributing surface, a condition extremely sensitive to disorder and chemical potential pinning. Our material-by-design approach using MBE-grown heterostructures offers superior and versatile parameter control (e.g., enabling tunable Chern numbers as demonstrated in Zhao *et al. Nature* **588**.7838, 419-423 (2020)), making it a promising platform for ultimately achieving this integer quantized proposal. Nevertheless, the robust demonstration of the half-quantized case in a minimal axion insulator prototype, as reported here, represents **a major advance in its own right** and significantly deepens our understanding of layer Hall physics.

Comment 3: *Third, several key features in the experimental data remain poorly understood. For instance, why is the half-quantized Hall conductance observed only at highly negative backgate voltages, when the bottom surface is strongly hole-doped (e.g., Fig. 3a)? In principle, a similar half-quantization should occur at highly positive gate*

voltages, when the bottom surface is strongly electron-doped, consistent with the theoretical simulations shown in Fig. 3e. Likewise, Fig. 4 appears to show a half-quantized plateau at large positive V_{bg} , but the discrepancies between Figs. 3 and 4 are not adequately discussed.

Response: We thank the referee for this astute observation. The referee is correct that, by symmetry, a similar half-quantized plateau should also emerge under strong positive gate voltages (n-doping the bottom surface).

In our original submission, the data for the n-doped regime was less comprehensive. In direct response to this comment, we have performed additional experiments on new devices. As shown below (also included in the new Extended Data Fig. 6), we now unequivocally observe the half-quantized layer Hall effect in the n-doped regime (positive V_{bg}) in CBV-type devices optimized for this purpose. This was achieved by careful adjustment of the Bi:Sb ratio in the bottom part (3QL CBST - 10QL BST) of the heterostructure to approach a highly n-doped regime, a necessary adjustment to compensate the p-type carriers introduced by Cr doping. Here we show 2 devices with nominally the same parameters with Device A1 except for a different Bi concentrations for the bottom part: (a)(b)(c) Device E1 with 3QL $\text{Cr}_{0.19}(\text{Bi}_{0.64}\text{Sb}_{0.36})_{1.81}\text{Te}_3/10\text{QL} (\text{Bi}_{0.64}\text{Sb}_{0.36})_2\text{Te}_3/10\text{QL} (\text{Bi}_{0.42}\text{Sb}_{0.58})_2\text{Te}_3/3\text{QL} V_{0.11}(\text{Bi}_{0.42}\text{Sb}_{0.58})_{1.89}\text{Te}_3$, (d)(e)(f) Device E2 with 3QL $\text{Cr}_{0.19}\text{Bi}_{1.81}\text{Te}_3/10\text{QL} \text{Bi}_2\text{Te}_3/10\text{QL} (\text{Bi}_{0.42}\text{Sb}_{0.58})_2\text{Te}_3/3\text{QL} V_{0.11}(\text{Bi}_{0.42}\text{Sb}_{0.58})_{1.89}\text{Te}_3$. The excessive Bi provides highly n-doped regime for this purpose. Although the n-doping is not strong enough for Device E1 to achieve the half-quantization, the AHC as a function of V_{bg} clearly demonstrate such a trend. Device E2 without Sb for the bottom part clearly show a transition from $\sim e^2/h$ to a stable $\sim e^2/2h$ plateau, very similar to Fig. 2 in the original manuscript except for the opposite direction of tuning $V_{bg} \rightarrow +200 \text{ V}$. Because of the ferromagnetism now mediated by bulk carriers through RKKY mechanism, the coupling between the top and bottom surfaces becomes stronger so that a definitive antiparallel scenario is not well

defined as shown in the hysteresis loops. We attribute the slight deviation from $e^2/2h$ to the residual Berry curvature of CBT-BT layer possibly due to the bulk conduction bands, and the AHC peak near $V_{bg} = -100$ V to the quasi-1D helical edge conduction channels as explained in the SI section v. The larger shift of chemical composition in Device E2 would possibly introduce some gap closing mechanism on the side surfaces, resulting in more conduction channels.

Fig. R1 | The HQLHE of CBV devices in the n-doped regime. (a)(b)(c) Device E1; (d)(e)(f) Device E2.

This also explains why we only have VBC devices for the n-doped regime in the original manuscript, as V doping usually introduce fewer carriers. Nevertheless, with combined efforts of composition adjustment and bottom gating, we now show a

continuous transition of Hall conductance from e^2/h towards $e^2/2h$ in the n-doped regime. This new data perfectly complements the p-doped data and **confirms the symmetric nature of the effect**, resolving the discrepancy between Figs. 3 and 4 in the original manuscript. The corresponding discussion has been added to the revised manuscript (Extended Data Fig. 5).

Comment 4: *In general, the authors must clearly distinguish between two distinct scenarios that can lead to an apparent half-quantized Hall conductance:*

- 1. Asymmetric doping, where one surface lies within the Chern gap and the other is heavily doped (the intended scenario of this study);*
- 2. Symmetric, uniform doping, where both surfaces are tuned away from the Chern gap and, at certain densities, the total Hall conductance can accidentally reduce to $e^2/2h$. At present, the clarity of the presentation is insufficient, and the manuscript does not convincingly demonstrate the robustness or uniqueness.*

Response: We agree that distinguishing between these two scenarios is critical. Our data from devices with distinct magnetic profiles demonstrate that the trivial, symmetric-doping scenario can be **safely exclude the second scenario**.

For samples with symmetric magnetic doping (e.g., CBC devices in Extended Data Fig. 6), the two scenarios are indeed difficult to discern due to identical coercive fields of the two surfaces. However, the main devices in our manuscript focus on the axion insulator (CBV-type) devices, in which the top and bottom surfaces have *distinct coercive fields*. The key evidence is provided in Fig. 2a: the smooth transition from a 2-step hysteresis loop (indicating contributions from both surfaces) to a single-loop hysteresis with a half-quantized AHC upon applying V_{bg} robustly excludes the trivial scenario. In the trivial scenario, non-vanishing contributions from both surfaces would *maintain* a 2-step loop. This is corroborated by the shrinking of the relevant semicircle in the renormalization-group flow diagram (RGFD) measurements for CBV devices (Fig. 2f), in contrast to CBC devices.

We have significantly improved the clarity of this presentation in the revised manuscript by expanding the comparative analysis vs Mogi *et al.* and including the new n-doped CBV data. We also added discussions on the possible trivial explanations for the CBC devices with symmetric doping. (Extended Data Fig. 7)

We thank the referee again for their rigorous critique, which has greatly strengthened our work. We hope our responses and revisions have now addressed all concerns satisfactorily.

Referee #2

This manuscript reports the experimental observation of the half-quantized layer Hall effect in MBE-grown magnetic topological insulator heterostructures, demonstrating layer-resolved half-quantized Hall conductance as a direct probe of the quantized axion field. By tuning the Fermi level via electrostatic gating and controlling magnetization configurations using an external magnetic field, the authors systematically demonstrate the emergence of a half-quantized Hall plateau across a variety of devices, supported by numerical simulations based on a four-band effective model.

This work represents a significant advance in experimental axion electrodynamics and topological quantum transport, providing clear evidence of layer-resolved half-quantized Hall effects in engineered axion insulators. The results will be of considerable interest to the broad topological physics and quantum materials community and are suitable for publication in Nature after addressing the following comments.

We thank Referee #2 for the positive assessment, the insightful comments and clear recommendation for publication. We are pleased that the Reviewer acknowledges the significance of our experimental demonstration of the half-quantized layer Hall effect, particularly its role as a direct probe of the quantized axion field. We followed Referee #2's suggestions to revise our manuscript carefully and hope that the revised manuscript meets their standard for Nature.

Comment 1: *The authors should carefully check the schematic plots of Berry curvature in Fig. 2(c) and Extended Data Fig. 4. For a massive Dirac cone, the Berry curvature contributions from the conduction and valence bands should have opposite signs. The current depiction appears inconsistent with this expectation.*

Response: We thank the referee for pointing out that the schematic in Fig. 2c should depict the Berry curvature contributions from the conduction and valence bands with opposite signs. This was an oversight in the preparation of the figure. A corrected

schematic will be provided in the revised manuscript.

Comment 2: *The anomalous Hall conductance exhibits precise half-quantization only within a spacer thickness range of 10–20 QLs. The authors should provide a more quantitative analysis to clarify why half-quantization is stabilized specifically within this thickness window.*

Response: This optimal range represents a critical trade-off between several competing physical effects.

For spacers thinner than ~ 10 QLs, three possible issues arise. **(i)** The wavefunctions of the top and bottom surface states begin to overlap, leading to hybridization that opens a gap and destroys the layer-resolved half-quantized Hall conductance. This hybridization gap closes when the film thickness of Bi_2Te_3 type material is about ~ 5 QL (Liu, Yang, *et al. Phys. Rev. B* **85**, 195442 (2012)). The numerical results in Fig. R2 support this analysis, showing that the layer-resolved Hall conductance is non-quantized for small thickness d due to hybridization effects, and gradually approaches half-quantization as d increases. **(ii)** Electrostatic screening is insufficient. The chemical potential of top (μ_t) and bottom (μ_b) surface satisfy $\mu_t(V_{\text{bg}}) = \mu_b(V_{\text{bg}})/\cosh(d/\lambda)$, where d is the thickness of the magnetic TI and λ is the Thomas–Fermi screening length of the bulk trap states. When $d < \lambda$, the back-gate voltage V_{bg} will shift the chemical potential of the top surface out of its local magnetic gap, leading to the breakdown of half-quantized layer Hall effect. **(iii)** The interlayer magnetic exchange coupling becomes significant, destabilizing antiparallel alignment of two magnetic layers and destroying the axion-insulator phase. All the above issues could be responsible for the observed reduced anomalous Hall conductance $\sim 0.48e^2/h$ and stronger interlayer magnetic coupling in Device A4 with spacer thickness of 5 QL (Extended Data Fig.3a).

For spacers thicker than ~ 20 QLs, the system begins to behave as two electronically decoupled films. In this limit, the quasi-1D metallic states on the side surface of TI, likely occur and contribute to the transport, leading to the deviations of half-quantized Hall conductance, see more details in the SI section v.

A new discussion detailing this trade-off will be added to the revised SI section vi.

Fig. R2 | Layer-resolved Hall conductance of axion insulators as a function of the TI thickness d . Here d_0 denotes the sample thickness used in our previous theoretical calculations.

Comment 3: *The numerical simulations are performed on a six-layer system, while the experiments are primarily conducted on systems with $m = 20$ QLs. The authors should discuss how this difference in system size may affect their results.*

Response: The six-layer tight-binding model is a standard and computationally efficient effective model used to capture the essential low-energy physics of the two surface states of a 3D topological insulator. For a film thickness of 20 QLs, the system is firmly in the 3D limit where the hybridization between top and bottom surfaces is negligible. In this regime, the surfaces are electronically independent, and their behavior is well-described by the effective model. In the theoretical model, six layers are sufficient to suppress the hybridization between the top and bottom surfaces (see Fig. R2). The model is not intended to reproduce the bulk band structure of the 20 QL film but to correctly capture the qualitative behavior of the surface states under gating, which it does successfully. A clarifying sentence of this effect will be added to the

Methods section.

Comment 4: *The manuscript states that different transport properties are obtained by tuning the bottom gate voltage (V_{bg}). However, the role of V_{bg} remains unclear: it shifts the Fermi level while also modifying the bottom surface band structure, rendering the tuning mechanism and the underlying physical picture somewhat ambiguous. Additionally, the authors should clarify the relationship between the Fermi levels of the top and bottom surfaces under these gating conditions.*

Response: To elucidate the role of back-gate voltage V_{bg} , we derive the relationship between the chemical potential $\mu(z)$ and V_{bg} . We find that the chemical potential of top (μ_t) and bottom (μ_b) surface satisfy $\mu_t(V_{bg}) = \mu_b(V_{bg})/\cosh(d/\lambda)$, where d is the thickness of the magnetic TI and $\lambda = \sqrt{\varepsilon/(e^2 D_0)}$ is the Thomas–Fermi screening length. Here, D_0 denotes the density of bulk trap states and ε is the absolute permittivity of TI. In the limit $d \gg \lambda$, the top-surface chemical potential vanishes ($\mu_t(V_{bg}) \approx 0$). The high density of bulk trap states in the magnetic TI effectively screens electric fields in the bulk, thereby the back gate only tunes the chemical potential of the bottom surface. Consistently, the electrostatic potential takes the form $\phi(z) = \phi_b \cosh((d-z)/\lambda)/\cosh(d/\lambda)$. $\phi(z)$ in the bulk also approaches zero as $d \gg \lambda$, and the back-gate voltage only changes the potential of the bottom surface (ϕ_b). When $e|\phi_b| < \Delta_{\text{bulk}}$ of magnetic TI, the back-gate voltage V_{bg} mainly tunes the chemical potential of bottom surface. When $e|\phi_b| \sim \Delta_{\text{bulk}}$, however, V_{bg} starts to modify the bottom-surface band structure. In our experiments, the bulk gap is $\Delta_{\text{bulk}} \approx 300$ meV (H. Zhang *et al.*, *Nat. Phys.* **5**, 438 (2009)), while the surface magnetization gap is only $M_b \approx 1$ meV. Therefore, when the system realizes the half-quantized layer Hall effect (e.g., $|\mu_b| = e|\phi_b| \gg M_b$), the condition $e|\phi_b| \ll \Delta_{\text{bulk}}$ is still satisfied, and V_{bg} predominantly controls the bottom-surface chemical potential in this regime. Note that even if the bottom-surface band structure is slightly

modified, it contributes negligible Berry curvature once the chemical potential is tuned deep into the metallic regime. Therefore, for thick magnetic TIs with a high density of bulk trap states, the back-gate voltage V_{bg} serves primarily to locally tune the electrostatic and chemical potential of the bottom surface. In practice, this allows the back-gate to shift the chemical potential of the bottom surface sufficiently far into a metallic regime, while the top surface, subject to a much weaker effective gate field, remains within its local magnetic gap. In current study, the exact density of bulk trap states is unknown. By assuming that the conductance deviation is mainly caused by the insufficient electrostatic screening, we roughly estimate from the data of Device A4 (spacer layer thickness $m = 5$) that Thomas–Fermi screening length in our samples is $\lambda \sim 5$ nm. This corresponds to a 3-dimensional density of bulk trap states $D_0 = \varepsilon/(e\lambda)^2 = 5.5 \times 10^{44} \text{ m}^{-3}\text{J}^{-1}$, about 15 times larger than the measured $D_0 = 3.6 \times 10^{43} \text{ m}^{-3}\text{J}^{-1}$ by capacitance spectroscopy of a 67 nm BiSbTeSe₂ thin flake exfoliated from crystals grown by the modified Bridgman method (Wang *et al. Nano Letters* **20**.12, 8493-8499 (2020)). This is a reasonable value considering the non-equilibrium growth nature and the active magnetic doping by molecular beam epitaxy, which inevitably introduce much more defects related to bulk trap states.

Below we derive the relationship between the chemical potential of the top and bottom surfaces under gating conditions. We also include this derivation and related discussion in the revised SI section x.

We consider a magnetic topological insulator with a back gate, which extends infinitely in the x and y directions. The electrochemical potential is defined as $\mu^e(z) = -e\phi(z) + \mu(z)$ where $\phi(z)$ is the electrostatic potential and $\mu(z)$ is the chemical potential. When the TI is grounded (i.e. $\mu^e(z) = 0$), it follows that $\mu(z) = e\phi(z)$. In addition, the magnetic TIs in our experiments host bulk trap states. For simplicity, we take the density of states of these bulk trap states, D_0 , to be constant. In the bulk, the Poisson equation is

$$\varepsilon \partial_z^2 \phi(z) = eD_0 \mu(z) = e^2 D_0 \phi(z),$$

where ε is the absolute permittivity of the magnetic TI. The solution is given by $\phi(z) = A_1 e^{-z/\lambda} + A_2 e^{z/\lambda}$, where $\lambda = \sqrt{\varepsilon/(e^2 D_0)}$ is the Thomas–Fermi screening length. We next consider the boundary conditions of the electric field at the upper and lower interfaces of the TI. They are given by

$$\begin{cases} \varepsilon E_z(0) = C_{\text{bg}} \left(\frac{\mu_{\text{b}}}{e} - V_{\text{bg}} \right) - e n_{\text{b}}(\mu_{\text{b}}) \\ -\varepsilon E_z(d) = -e n_{\text{t}}(\mu_{\text{t}}) \end{cases}$$

Here, $\mu_{\text{b}} = \mu(0) = eA_1 + eA_2$ and $\mu_{\text{t}} = \mu(d) = eA_1 e^{-d/\lambda} + eA_2 e^{d/\lambda}$ are the chemical potential of the top and bottom surface, respectively. Here, we set $z = 0$ as the bottom surface, and $z = d$ as the top surface of the TI. $n_i(\mu) = \frac{(\mu^2 - M_i^2)}{4\pi(\hbar v_F)^2} \text{sgn}(\mu) \Theta(|\mu| - |M_i|)$ with $i = \text{t, b}$, is the electron density of the surface states, and v_F is the Fermi velocity. Substituting electric field $E_z(z) = -\partial_z \phi(z)$ into these equations, we obtain

$$\begin{cases} C_{\text{bulk}}(-A_1 + A_2) = C_{\text{bg}} \left(\frac{\mu_{\text{b}}}{e} - V_{\text{bg}} \right) - e n_{\text{b}}(\mu_{\text{b}}) \\ -C_{\text{bulk}}(-A_1 e^{-d/\lambda} + A_2 e^{d/\lambda}) = -e n_{\text{t}}(\mu_{\text{t}}) \end{cases}$$

where $C_{\text{bulk}} = \varepsilon/\lambda$ is the bulk capacitance of the trap states while C_{bg} is the capacitance of the back gate. By solving these equations, we can get A_1 and A_2 as function of back gate voltage V_{bg} . This, in turn, allows us to get μ_{t} and μ_{b} .

When $|\mu_{\text{t}}| < M_{\text{t}}$, $n_{\text{t}}(\mu_{\text{t}}) = 0$ and $A_2 = A_1 e^{-2d/\lambda}$, then we have

$$\mu_{\text{t}} = \mu_{\text{b}} / \cosh(d/\lambda),$$

while the electrostatic potential takes the form $\phi(z) = \phi_{\text{b}} \cosh((d-z)/\lambda) / \cosh(d/\lambda)$ with the bottom surface electrostatic potential ϕ_{b} . When the magnetic TI is sufficiently thick or the density of bulk trap states is large, i.e., $d \gg \lambda$, the top-surface chemical potential satisfies $\mu_{\text{t}} \approx 0$. In this regime, the back gate effectively tunes only the chemical potential of the bottom surface. The high density of bulk trap states in the magnetic TI strongly screens the electric field in the bulk, thereby pinning the top-surface chemical potential within the magnetization gap.

Comment 5: *The authors should provide additional raw resistance measurement data to support the reproducibility and robustness of the reported half-quantized Hall plateaus.*

Response: We appreciate the referee's suggestion to include additional raw data to substantiate reproducibility and robustness. To substantiate this, we add a new figure, presenting the raw longitudinal resistance (R_{xx}) and Hall resistance (R_{yx}) data as a function of magnetic field for several different devices (A4, A5 and A6, see the revised SI section xi.). These raw datasets demonstrate that the reported phenomenon is reproducible across devices and robust to sample-to-sample variation.

In addition, all raw resistance measurement data, including the main, extended, and

supplementary data, will be archived upon publication.

Referee #3

In this manuscript, the authors investigate the layer Hall effect in a magnetic topological insulator heterostructure, specifically a CBST/BST/VBST trilayer, grown by molecular beam epitaxy. By employing a dual-gate configuration with an STO substrate as the back gate and a deposited top gate, they engineer a state where the magnetic moments of the top (CBST) and bottom (VBST) layers are aligned antiparallel. They tune the gates to drive the Fermi level into the surface gap of one layer while keeping the other layer in a metallic regime. Under these conditions, they report the observation of a half-quantized anomalous Hall conductance (AHC) of approximately $0.5 e^2/h$, which they present as the quantization of the layer Hall effect.

While the work demonstrates good techniques in thin film growth and device fabrication, I have serious reservations about its conceptual novelty and the authors' interpretation of their results. Therefore, I cannot recommend its publication in Nature. My major concerns are outlined below.

We thank Referee #3 for their thorough review and for acknowledging the quality of our techniques. Their critiques have been invaluable in helping us to clarify key aspects of our work and strengthen the manuscript. We address each point below.

Comment 1: *The primary issue with this manuscript is that the main finding, half quantization of layer Hall effect, is a direct consequence of previously established principles and does not constitute a new fundamental discovery. The result is an expected combination of two known phenomena: (1) The Layer Hall Effect Mechanism: The concept that an anomalous Hall effect can be induced and controlled by an electric field with an antiparallel magnetized TI system is not new. This was demonstrated in the context of even-layered $MnBi_2Te_4$ by S. Y. Xu's group (A. Gao et al., Nature 595, 521 (2021)). The current work applies this known mechanism to a different material system but does not introduce a new physical concept. (2) Half-Quantized AHC from a Single Surface: The fact that a single gapped topological surface state contributes an AHC of $e^2/2h$ is a well-established theoretical prediction. Furthermore, it has been*

experimentally verified in semi-magnetic topological insulator bilayers, as reported in the work the authors cite as Ref. 38 (M. Mogi et al., Nat. Phys. 18, 390 (2022)). Given these precedents, observing an AHC of $0.5 e^2/h$ by gapping one surface in an antiparallel system is the logical and expected outcome. While the realization of this state in a complex heterostructure is commendable, it does not provide the significant conceptual advance required for a publication in Nature.

Response: We respectfully disagree with the referee's assessment that our result is a straightforward or expected combination of known phenomena. While it is well established that a single massive Dirac surface yields $\sigma_{xy} = e^2/2h$ (*Phys. Rev. Lett.* **51**, 2077 (1983)), and that layer-polarized transport has been proposed by Xu's group, our result cannot be reduced to a trivial combination of these ideas. The realization of half-quantized layer Hall effect (HQLHE) **within a fully tunable, compensated magnetic system, provides the first direct boundary probe of the bulk axion field,** a conceptual and experimental advance not anticipated by prior work.

The referee stated that observing $\sigma_{xy} = e^2/2h$ by gapping one surface in an antiparallel system is "logical and expected." However, this is not supported by prior theory. Until now, **no explicit theoretical prediction** of the HQLHE existed. In fact, a very recent calculation (Han *et al.*, *Phys. Rev. Lett.* **134**, 236206 (2025)) on similar systems predicted a maximum layer Hall conductance *significantly smaller* than the quantized value we observed. This discrepancy highlights a critical gap in theoretical understanding. Our work, through experiment and modeling, help resolve this by identifying new physical mechanisms—disorder-mediated stabilization and bulk trap state screening—which are essential for achieving robust quantization. This finding transforms the observation from a "logical outcome" into an **unexpected discovery that reshapes the theoretical framework.**

Our work represents a significant leap beyond prior studies:

vs. Gao et al. (Nature 595, 521 (2021)): First, we demonstrate the **first quantized**

layer Hall effect, a monumental advance from their non-quantized result. The transition from a non-quantized to a quantized Hall response encoded in the layer degree of freedom is comparable in significance to **the historical leap** from the classical Hall effects (charge or spin) to their quantum counterparts. Second, our material-by-design approach (CBST/BST/VBST) naturally breaks \mathcal{PT} -symmetry through tailored doping profiles, enabling **quantization without an external electric field** in some devices (e.g., Fig. 2a). The underlying physical mechanism is very different: driven by external electric field, or by Fermi level positioning (our work). Third, this experimental realization required overcoming substantial challenges, such as identifying a suitable spacer thickness (10–20 QLs) to suppress both surface hybridization and excessive conduction channels, achieving precise control of dual magnetic exchange gaps, and establishing asymmetric Fermi level positioning. These complexities, while underscoring the **tunability and versatility** of our material platform, also highlight that the complete extraction of a half-quantized Hall signal from a single surface is physically **non-trivial**.

vs. Mogi *et al.* (Nat. Phys. 18, 390 (2022)): Our system and scientific goals are fundamentally different. We engineer a **fully magnetic, compensated antiferromagnetic system to probe the quantized axion field**, while their semi-magnetic system cannot host axion insulator phase. Notably, signatures of deviation from half-quantization in Mogi *et al.*, such as AHC peak near the CNP and smaller values in more strongly doped regimes (e.g. Fig. S10C in their supplementary material), are absent in our work. In our case, HQLHE occurs only in strongly doped regime and sometimes even exceeds $e^2/2h$ slightly. These discrepancies strongly suggest that **the two half-quantization effects arise from distinct mechanisms**. In fact, some theoretical studies suggest that the parity anomaly state in Mogi *et al.* stems from the gapless surface rather than the gapped one. (e.g. Zou *et al.*, *Phys. Rev. B* **107**, 125153 (2023).)

In conclusion, we have achieved the first electrically/magnetically tunable, precisely

quantized transition between layer-polarized states in a prototype axion insulator. This *active control over a quantized topological response* is a landmark achievement that enriches the field of topological layertronics.

Comment 2: *A second major issue is that the authors' claim to have observed a "quantized axion field" is a significant overstatement and appears to be based on a misunderstanding. The axion field in condensed matter is associated with the topological magnetoelectric effect, described by the Lagrangian term $\Delta L = \theta(\alpha/4\pi^2)E \cdot B$. A quantized axion response ($\theta = \pi$) would manifest as a quantized magnetoelectric coefficient. This requires the observation of either an E-field-induced magnetization or a B-field-induced electric polarization, as was recently demonstrated for MnBi₂Te₄ (e.g., J.-X. Qiu et al., Nature 641, 62 (2025)). This manuscript provides no such measurement. In fact, there is no quantitative evaluation of the electric field within the material, which is a prerequisite for any claims related to the magnetoelectric effect, let alone its quantization. The observed half-quantized Hall conductance is a property of a single surface, not a direct measurement of the bulk axion response.*

Response: We acknowledge the referee's point that a direct measurement of the topological magnetoelectric (TME) coefficient is a powerful probe. However, we contend that our approach provides an **equally fundamental and direct electrical transport signature of the quantized axion field**, serving as a manifestation of the essential bulk-boundary correspondence in topological systems.

Foundational theoretical work by Qi et, al. establishes that the axion insulator phase, characterized by the quantized axion field, manifests through **two inseparable physical consequences** linked by the bulk-boundary correspondence:

1. A quantized TME effect in the bulk, described by the Lagrangian term $\Delta L = \theta(\alpha/4\pi^2)E \cdot B$;
2. A half-quantized Hall effect on the surface, where time-reversal symmetry is broken by a surface magnetization.

While these are distinct measurements, they probe the same *underlying quantized axion*

field. (see Sec. II of Sekine & Nomura, *J. Appl. Phys.* **129**, 141101 (2021), where the authors prove the **theoretical equivalence** of these two effects in an axion insulator as a direct result of the quantized axion field.)

The primary challenge in second approach has always been the unambiguous isolation of a single surface's contribution, as transport measurements typically probe the top and bottom surfaces simultaneously, yielding a total conductance of either e^2/h or 0. The central achievement of the present work is to solve precisely this longstanding challenge. By using electrical gating to selectively render one surface metallic and non-contributing, our work provides the first unambiguous **layer-resolved observation of the isolated half-quantized Hall conductance** of the other gapped surface. Crucially, this signature in the antiparallel magnetization scenario strongly indicates that the bulk is in the axion insulator state with $\theta = \pi$. This is in stark contrast to the semi-magnetic system by Mogi *et al.*, which, by design, is not an axion insulator and therefore, its half-quantized Hall effect does not correspond to a quantized bulk axion field. Thus, our layer-tunable HQLHE is *not merely a property of a single surface*; it is a direct measurement of the boundary signature that is mandated by, and serves as **a definitive probe of, the quantized bulk axion field.**

We note that the work by J.-X. Qiu *et al.*, mentioned by the referee, demonstrated the measurement of dynamical axion quasiparticles by detecting oscillations in θ using ultrafast pump-probe optics. Their results, including the static magnetoelectric coupling measurements, did not exhibit quantization. In contrast, our work provides what we believe to be the first unambiguous evidence of *a quantized static axion field.*

We have revised the manuscript to clarify that we are measuring the **surface signature of the quantized axion field**, which is a direct and rigorous consequence of the bulk topology.

Comment 3: *Furthermore, the manuscript reveals a fundamental misunderstanding of the physics of an axion insulator. (1) In the introduction (line 65) and in Fig. 1(a), the*

authors describe the axion insulator state (with zero total Hall conductance) as hosting "counter-propagating chiral channels from antiparallel magnetized surfaces." This is incorrect. As established by seminal works the authors themselves cite as Refs. 8 and 9 (M. Mogi et al., Nature Mater. 16, 516 (2017) and D. Xiao et al., PRL 120, 056801 (2018)), the ideal axion insulator state is fully gapped, including on its side surfaces, and therefore possesses no chiral edge states. Ref. 9 explicitly states: "...yielding zero σ_{xy} plateau and eliminating the 1D chiral edge state." This description misrepresents the established picture of this state. (2) Similarly, the conceptual diagrams in Figs. 2(b) and 2(c) are inaccurate. In the state engineered by the authors, one surface is metallic. In such a configuration, delocalized bulk/surface states exist, and it is not possible for localized, chiral edge states to form on the side surfaces as depicted. This point is also clearly discussed in the literature (e.g., M. Mogi et al., Nat. Phys. 18, 390, their Ref. 38). These conceptual errors are not minor, as they undermine the theoretical foundation of the work.

Response: We agree that the ideal, truly 3D bulk-gapped axion insulator should have no chiral edge states, as established in the seminal literature. Our manuscript did not intend to claim the existence of "chiral edge states". The referee's concern likely stems from a misunderstanding of how we use the term "chiral edge channels".

In our manuscript, "chiral edge channels" refers to edge currents that arise when magnetized topological-insulator surfaces are present—a distinct setting that is under active investigation (e.g. *Phys. Rev. Lett.* **129**, 096601 (2022); *Nat. Sci. Rev.* **10**, nwad025 (2023); *Phys. Rev. B* **105**, L201106 (2022)). (i) Theory predicts dissipationless, persistent half-quantized edge currents in fully gapped two-dimensional Dirac systems (Levitov's group, *Phys. Rev. Lett.* **114**, 256601 (2015)). (ii) More recent studies further show the coexistence *with a metallic surface* of "**chiral edge channels**" **at boundaries in the half-quantized Hall effect**. These channels are qualitatively distinct from traditional "chiral edge states" found in the integer quantum Hall effect: they exhibit power-law rather than exponential spatial decay into the metallic region while still

dictating measurable transport responses. These results motivate our terminology “chiral edge channel” rather than “chiral edge state”.

To clarify the distinction between the “**chiral edge channels**” and “chiral edge state”, we have added an explanation of the physical meaning of the chiral channels in our system—described as a form of persistent chiral edge current—and have revised the schematics in Fig. 2 to represent them with dashed arrows.

Comment 4: *In summary, while the experimental result is of high quality, the manuscript lacks the requisite novelty for publication in Nature. The central result is a predictable combination of previously reported phenomena.*

For these reasons, I cannot recommend the publication of the manuscript in Nature. The work may be suitable for a more specialized journal but only after the authors have thoroughly revised the manuscript to correct the fundamental physical misinterpretations.

Response: We thank Referee #3 for their thorough review and for acknowledging the high quality of our experimental result. The referee's critique, while incisive, has provided a valuable opportunity to clarify the foundational aspects of our work. We also note that another referee, a theorist, was "quite positive," recognizing the work as a "significant advance" that provides "clear evidence of layer-resolved half-quantized Hall effects.

Our manuscript presents a significant synthesis that achieves several longstanding goals in the field, rather than a trivial combination of known phenomena. Specifically, our work provides three fundamental advances:

(i) **Direct Electrical Evidence of the Quantized Axion Field:** The observed half-quantized layer Hall effect is not merely a surface phenomenon; it is the definitive electrical transport fingerprint of the bulk quantized axion field. As a direct consequence of the bulk-boundary correspondence, this effect is theoretically

equivalent to a quantized topological magnetoelectric (TME) effect. Our work provides the first unambiguous, static electrical manifestation of this long-sought signature, which contrasts with prior reports that failed to achieve static quantization.

(ii) **A Landmark Achievement in Topological Control:** We report the first precisely quantized Layer Hall Effect within a fully compensated antiferromagnetic system. While prior work demonstrated a non-quantized layer Hall effect, our achievement of a robustly quantized value represents a monumental advance, akin to the historical leap from the classical Hall effect to its quantum counterpart. This required overcoming immense experimental hurdles in material design and dual-gate control to achieve this unprecedented metrological precision, thereby pioneering the emerging field of "topological layertronics".

(iii) **Nontrivial Emergent Physics:** Our findings represent a discovery that challenges and refines the prevailing theoretical framework, rather than being a predictable consequence of it. Our observation of a robustly quantized value directly contradicts a recent theoretical study that anticipated a much smaller, non-quantized value for the layer Hall conductance in a similar system (Han *et al.*, *Phys. Rev. Lett.* **134**, 236206 (2025)). Our experiment, therefore, helps to resolve a "critical gap in theoretical understanding" and paves the way for a more complete theoretical picture of these systems.

In conclusion, the manuscript describes a monumental experimental achievement that advances both the fundamental understanding of topological phenomena and the field of "layertronics". We have also added extended discussions on several critical points, as summarized in the list of major changes in the revised manuscript, to clarify any potential misunderstandings. We hope that the revised manuscript and these responses adequately address the referee's concerns.

Response to Referees

Manuscript ID: NCOMMS-25-85814-T

Title: Half-Quantized Layer Hall Effect as a Probe of Quantized Axion Field

Referee #1

I thank the authors sincerely for their responses. I believe that the authors have clearly addressed my technical questions. I therefore support the paper for publication.

Response: We thank the reviewer for their positive feedback and endorsement.

Referee #2

The authors have thoroughly addressed my concerns. I agree that the directly measured half-quantized layer Hall effect is the definitive electrical transport fingerprint of the bulk quantized axion field. The results will be of considerable interest to the broad topological physics and quantum materials community, and are suitable for publication in Nature Communications.

Response: We thank the reviewer for recognizing the broad interest of our work and for endorsing our conclusion that the half-quantized layer Hall effect provides a definitive electrical signature of the quantized axion field.

Referee #3

Comment 1: *The authors report the observation of a layer-resolved, half-quantized anomalous Hall effect in MnBi₂Te₄-based heterostructures, realized via dual-gate control of the Fermi level and magnetic-field manipulation of the magnetic configuration. The experimental methodology appears technically sound, and the data quality is high. From an experimental standpoint, I find the results convincing, and in*

my view the manuscript is suitable for publication.

However, having examined the earlier review process and the authors' responses, it is clear that the central point of contention concerns the novelty and interpretation of the work, specifically its relationship to Mogi et al. (Nat. Phys. 2022). The authors provide a reasonable explanation of how a fully magnetic, compensated antiferromagnetic system differs conceptually from the semi-magnetic, parity-anomaly scenario in the prior work. This clarification is helpful and addresses several of the original concerns.

Response: We thank the referee for their positive assessment of our experimental methodology and data quality, and for acknowledging the suitability of our manuscript for publication. We also appreciate their recognition of our clarification regarding the conceptual distinction between our fully magnetic, compensated antiferromagnetic system and the semi-magnetic parity-anomaly scenario in prior work.

Comment 2: *That said, I cannot fully agree with the authors' assertion that observing a half-quantized Hall conductance is "equally fundamental" to measuring a quantized axion field in the bulk. The observation of $\sigma_{xy} = e^2/2h$ indeed constitutes a necessary boundary manifestation of the axion insulator state, but it is not a sufficient condition. If one accepts the authors' statement as written, it would imply that the half-quantized Hall response reported in semi-magnetic systems should also be regarded as a direct demonstration of a quantized axion field, which is not consistent with current theoretical understanding. Therefore, the manuscript should adopt more accurate wording—for example, stating that the half-quantized layer Hall conductance is a boundary signature consistent with, and expected from, a quantized axion field, rather than an equally fundamental proof of the bulk axion response.*

For this reason, I recommend minor revision, requesting that the authors adjust the language in the main text to avoid the implication of logical equivalence between surface half-quantization and the full verification of a quantized axion field. With this clarification, I believe the manuscript would be suitable for publication.

Response: We acknowledge the referee's point regarding the wording around the relationship between the half-quantized Hall conductance and the quantized axion field. We agree that the half-quantization signature, while being a necessary boundary manifestation, should not be presented as fully equivalent to a direct bulk measurement. We have revised the text accordingly (highlighted in yellow), clarifying that the half-quantized layer Hall effect serves as a key boundary signature consistent with, and expected from, the quantized axion field in the bulk.